# Towards Multi-view, Explainable and Scalable Representation Learning for Spoofed Audio Detection

## Abstract

Artificial Intelligence (AI)-generated content (deepfake content) is considered to be a major threat that can lead to fraud and the spread of incorrect information. The focus of generative AI research has largely been on advancements in content generation, with less attention given to detection of AI generated content, particularly for AI generated audio content. Although foundation models offer powerful representations for detecting spoofed audio, they suffer from limitations in explainability and slow extraction speeds, which affect scalability. Prior research has integrated sociolinguistic expertise to identify phonetic and phonological cues in spoken English for spoofed audio detection. This approach, while successful, was limited in scale as it relied on the labeling of linguistic features by domain experts. In this paper we propose a novel model to auto-label expert-informed phonetic and phonological cues through deep-learning based representations fine tuned with domain expert input. As such, using the fine-tuning method with expert-informed features, we scale this interdisciplinary approach and demonstrate its benefits in enhancing explainability and optimizing resource utilization (consumed time) for utilizing foundation models in large-scale applications. For example, when considering XLSR-Wav2vec-ResNet18 as one of the most recent baselines, findings indicate that our method has decreased the Equal Error Rate of this baseline model in audio deepfake detection with at least 7% (effectiveness). In our proposed cost efficient ensemble setup, we have 31% time reduction in audio deepfake detection (scalability). Additionally, the algorithmically encoded linguistic features enhance the explainability via reverse engineering (explainability). Our proposed method is a multi-view approach as it takes advantage of not only deep representations, but also human expert-informed phonetic and phonological aspects of natural speech.

## Introduction

With the rapid progress of generative AI, producing highly realistic fake content has become both quick and effortless Khanjani et al. (2022). Among various forms of synthetic media, fake audio is especially concerning due to its potential to disseminate misinformation and facilitate fraudulent actions. Although AI generated voices can have positive applications, like voice reconstruction for people with advanced motor neuron disease who can no longer speak, there are many recent cases of deepfakes as threats to people's privacy and online security, such as: Stupp (2019); Nikki Main (2023); Smith (2021). For example, AI-generated robocalls have been utilized to cause confusion in political elections, solely through audio messages Verma & Kornfield (2024). Thus, there is strong motivation to advance Audio Deepfake Detection (ADD). Despite recent progress, baseline ADD models continue to struggle with generalizability, explainability, and detection speed, affecting their scalability potential. Representation learning is a key approach to addressing these challenges, as more informative input features can improve classification performance, enhance explainability, and directly reduce detection time.

Different representations are used for ADD including: 1) *Hand-crafted acoustic representations:* Researchers utilize hand-crafted audio acoustic features, which are used to train deep learning models for binary classification tasks. While this approach is very common in ADD, a recent study shows these features' performance drops significantly when faced with out-of-domain datasets Yang et al.

(2024), and their generalizability is under question. Another recent study Zhang et al. (2025a) creates a multi-view approach across different acoustic features to enhance their ability to capture fake audio; however, their approach is not evaluated against popular speech foundation-based models such as Yang et al. (2024). 2) *Deep Neural Network (DNN)-based Features and foundation models*: offers another type of representation that has shown strong ability in ADD. DNN-based features can be trained with fake and real audio data (such as RawNet2 end to end model Tak et al. (2021)) or using pre-trained, usually self supervised large DNNs (foundation models), to obtain the final representation (such as Zhang et al. (2023); Tak et al. (2022)). DNN based features' performance drop is much less than in the case of acoustic features Yang et al. (2024) when faced with an out-of-domain dataset. According to Yang et al. (2024), three pretrained models that had the least performance drop and most generalizability are: Wav2Vec-XLSR Babu et al. (2021)[1] (pre-trained on 128 languages), HuBERT Hsu et al. (2021) (pre-trained on LibriSpeech Panayotov et al. (2015)) and WavLM Chen et al. (2022) (trained on Libri-Light, GigaSpeech, and VoxPopuli datasets). These representations have great potential to act as the most recent baselines for recently developed ADD models. Whisper is another common audio pretrained DNN based representation, developed by OpenAI Radford et al. (2023). While Whisper model works well in not out-of-domain datasets Pham et al. (2024), in comparison of generalizability, it is not as successful as the other mentioned foundation models, as shown by results in Yang et al. (2024), therefore, we do not consider this model as one of the baselines. 3) *Expert-in-the-loop Representations:* Recent research has demonstrated novel representations for detecting spoofed audio. A study utilized principles from articulatory phonetics and fluid dynamics to determine whether a speech sample in question could originate from the human vocal tract Blue et al. (2022). Yet, this approach frequently involves extracting particular phoneme sets tailored to each dataset, which makes it time-intensive and limits its generalizability Zhang et al. (2025b). Moreover, it often emphasizes individual phonemes while overlooking the temporal structure of the entire phoneme sequence Zhang et al. (2025b). Another study Zhang et al. (2025b) tried to fill the gaps in the aforementioned research "by focusing on phoneme-level speech features" . They converted the frame level attributes to a sequence of phoneme level features. They demonstrated that "inconsistencies in phoneme-level features between real and fake samples" can be a distinguishing characteristic for ADD Zhang et al. (2025b). A recent approach Zahra Khanjani (2023) demonstrated how incorporating manually extracted phonetic features can enhance the standard baselines established by the ASVspoof 2021 Challenge Yamagishi et al. (2021) across a hybrid dataset (containing multi-type of attacks). These studies introduced Expert Defined Linguistic Features (EDLFs), which are linguistic features selected by sociolinguists with domain knowledge of features unique to human speech. EDLFs were integrated into the training datasets used to develop various machine learning models aimed at ADD. Their findings indicated that a Logistic Regression (LR) model trained on EDLFs surpassed multiple baseline systems from the ASVspoof 2021 challenge Zahra Khanjani (2023); Yamagishi et al. (2021); Anonymous (2024) across their hybrid dataset. More recent studies Boumber et al. (2024); Khanjani et al. (2025); Mallinson et al. (2024) have demonstrated the strong potential of multidisciplinary approaches and expert-in-the-loop representations in the field of deception detection and combating spoofed audio.

Although audio deepfake detection has been a hot topic recently, ADD models still struggle with the generalization issue AlAli & Theodorakopoulos (2023); Pham et al. (2024). Those models that have been shown to be generalizable tend to suffer from a lack of explainability and scalability. Their ability to be used as a real-time scalable approach is still under question and has not been investigated adequately for ADD. Most foundation models require a time-consuming feature extraction phase threatening the capacity for scalability crucial for detection at scale. In the current study, inspired by Zahra Khanjani (2023), we leverage the potential of expert-in-the-loop representations in audio deepfake detection as one of the views in our proposed multi-view approach. We demonstrate how this multi-view approach can fill the gap (explainability and scalability) while maintaining models' effectiveness by fine tuning based on domain experts' inputs to foundation models. Our contributions are as follows:

- *Designing an explainable expert-in-the-loop representation learning approach:* Considering the best-performing foundation models (Wavw2vec-XLSR, HuBERT and WavLM) as our baselines, we demonstrate how we can add explainability to the deep models through our expert-in-the-loop representations and reverse engineering to discover the features that are indicative of deepfake behavior. We call our expert-in-the-loop representation learning

---

[1]https://huggingface.co/facebook/wav2vec2-xls-r-300m

approach *ALiRAS: Auto-labeled Linguistic Representations for Audio Spoofing detection*. ALiRAS provides explainability while detecting spoofed audio. To the best of our knowledge, this research is the first work on explainability of ADD models using an expert-in-the-loop approach.

- *Scalable solution:* We question the foundation models used in terms of consumed resources and the potential to be utilized in real-time scenarios at scale. We show ALiRAS can help in saving resources through the proposed cost-efficient ensemble method that led to a 31% reduction in the consumed time in our experiments. To the best of our knowledge, this study is the first in ADD investigating the detectors based on consumed time, while leveraging the benefits of the existing foundation models. We also make the previously introduced linguistic approach scalable through the linguistic auto-labeling process.

- *Effectiveness:* In addition to explainability and resource efficiency, the ALiRAS model when combined with the bseline foundation models has shown maintained ADD performance (in terms of ROC AUC and EER).

We call our ALiRAS approach a multi-view representation learning as we utilize multiple views in ADD: a baseline model view and the phonetic and phonological linguistic view. The rest of the paper is organized as follows. Section  discusses the problem formulation as well as the overall methodology. In section  we demonstrate the dataset, experimental details and results, then we continue to the conclusion and future steps in section .

## METHODS

Figure 1 illustrates our overall methodology. Audio datasets were provided to linguistic experts to build upon the approach introduced in Zahra Khanjani (2023). Our linguistic team employed the same linguistic labeling methodology described in Zahra Khanjani (2023) to extract phonetic and phonological features that differentiate between genuine and spoofed audio. These features were subsequently used to train machine learning models via the auto-labeling module, as depicted in Figure 1. The auto-labeling module generates the Auto-labeled Linguistic Representations for Audio Spoofing detection (ALiRAS), which are then integrated into the ADD system. These features serve as inputs to the ADD system augmenting the existing popular representations such as foundation models and bringing the advantage of explainability through reverse engineering. Therefore, for each audio that is labeled spoofed, we know the auto-labeled linguistic features explaining why this label is chosen by the classifier, which are based on the expert-in-the-loop representations, and provides us with semantic meaning of the classifier's decision. We also show how this model augmentation saves resources compared to foundation models alone, which are resource carving and time-consuming. In our proposed cost-efficient ensemble method, we use ALiRAS-MLP as the first layer of classification. Instead of passing the whole dataset, only the audio clips that are not labeled as spoofed by the ALiRAS-based model (true and false negative samples) continue to the second layer of classification which is one of the baselines (XLSR-ResNet18, HuBERT-ResNet18, and WavLM-MLP). In our dataset, this method led to processing 31% less data for the time-consuming extraction of the large foundation models.

We next describe each step demonstrated in Figure 1.

**Experts' Extracted Representations ($m_r$):** Three linguistic features were extracted by sociolinguistic experts, following the methodology outlined in Zahra Khanjani (2023). These features include: (1) audible intake or outtake of breath (presence = 1, absence = 0), (2) anomalous pitch production (true = 1, false = 0), and (3) anomalous audio quality (true = 1, false = 0). Equation 1 defines the extracted linguistic features. The details of the linguistic labeling process are consistent with prior work in this domain Zahra Khanjani (2023).

*Definition 1:* Given an audio clip $a_i$ , with the linguistic feature set where:

$$a_i^{\text{linguistic features}} = [a_i^{\text{presence-of-breath}}, a_i^{\text{pitch-anomaly}}, a_i^{\text{audio-quality-anomaly}}] \tag{1}$$

The feature $a_i^{presence-of-breath}$ means that experts identified an audible presence of breath in an audio. For anomalous pitch ($a_i^{pitch-anomaly}$), if sociolinguistic experts identified any instances where the pitch appeared anomalous—such as being markedly higher or lower than expected, or exhibiting irregular fluctuations—the sample was labeled with a value of 1. Conversely, samples in which the pitch was consistently perceived as typical and within the expected range of variation for spoken

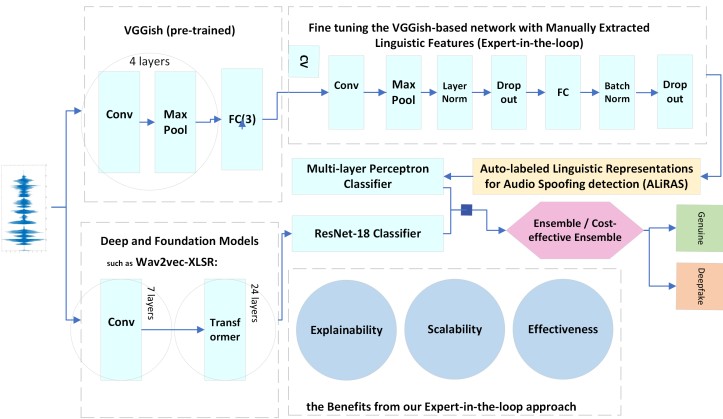

Figure 1: **Overall Methodology to utilize our proposed expert-in-the-loop representations in ADD.** CV: Cross Validation, Conv: Convolutional layer. FC: Fully connected layer. ResNet: Residual Neural Network.

English were assigned a value of 0 Zahra Khanjani (2023). For audio quality, a sample is labeled as 1 if the experts perceived any anomalies in the overall audio quality—such as distortion, compression, or characteristics like tinny or robotic-sounding audio. In contrast, samples with audio quality that was consistently judged to be typical and within the expected range of natural English speech were assigned a label of 0 Zahra Khanjani (2023). Various classification algorithms were trained using these linguistic features for the task of ADD, including Support Vector Machines (SVM), LR, Random Forests (RF), and Multi-layer Perceptrons (MLP). Among these, LR demonstrated the most notable performance in ADD when fed by the manually extracted linguistic features.

**Auto-labeling:** The demands of large-scale deployment highlight the importance of developing auto-labeling processes that leverage AI models while still incorporating domain expert input for effective model training and validation. Prior work has addressed the role of breath features in spoofed audio detection, such as the Breathing-Talking-Silence encoder Doan et al. (2023). However, these approaches did not involve collaboration with linguistic experts, limiting the depth of linguistic feature analysis and potential extensions Anonymous (2024). Furthermore, these studies did not evaluate the impact of their methods on resource efficiency when integrated with standard baseline systems, leaving the benefits of the interdisciplinary approach largely unexplored. We created an automated process to achieve ALiRAS, which is designed to use AI models for labeling data with linguistic features for ADD. We fine tuned classification algorithms (SVM, Convolutional Neural Networks (CNN), LR, MLP, RF, and XGboost) using the experts' extracted representations. Also, various embeddings were used to auto-label linguistic features including: *a) Acoustic Features:* sharpness, loudness, average roughness of audio, mel-spectrograms, and Mel Frequency Cepstral Coefficients (MFCCs); this group of features failed significantly in capturing the aforementioned linguistic features in our setup. *b) Deep and Foundation Features:* foundation models such as Hu-BERT Hsu et al. (2021), WavLM Chen et al. (2022), Wav2Vec-XLSR Babu et al. (2021), and a deep CNN-based model called VGGish Hershey et al. (2017) are also evaluated to auto-label the linguistic features as the results are demonstrated in the section . We also experimented with multiple setups including: *a: Both multi-label classification and binary setup were tried; with the better performance seen for binary classification for the purpose of auto-labeling the linguistic features. b: For large embeddings, both encoded (to 64 dimension) and non-encoded were tried, and not much difference in performance was captured.*

Based on the described ALiRAS methodology, we pre-trained VGGish-based models (better than other input features in auto-labeling Receiver Operating Characteristic - Area Under the Curve - ROC AUC) using the experts' extracted representations, and developed three binary classifiers, each dedicated to automatically labeling one of the three specified linguistic features. These pre-trained models are relatively lightweight, making them computationally efficient and suitable for deployment on CPU resources with minimal overhead. The final classifier algorithm is CNN for all of the aforementioned models. Each auto-labeling model contains the following number of parameters: 1,442,305 for detecting breath presence, 721,217 for identifying pitch anomalies, and 14,817 for detecting audio quality anomalies.

**Baselines and ensemble Modeling:** We leverage the full advantages of ALiRAS by integrating it into ensemble models alongside established baseline systems. As our primary baseline, we adopt the three top-performing models identified in the literature Yang et al. (2024). One baseline employs Wav2Vec-XLSR embeddings as input to a ResNet18 classifier. Another uses HuBERT representations with the same ResNet18 architecture. The third baseline leverages WavLM representations fed into an MLP classifier, which demonstrated superior detection performance compared to ResNet18 for WavLM. We demonstrate that incorporating ALiRAS into ensemble with these baselines not only maintains performance but also results in resource/time savings. Additionally, we are able to improve the explainability of these methodsThe ensemble configurations evaluated in our study are as follows: a) We use each type of representation separately (ALiRAS and baseline or deep feature). Then, we create an ensemble model based on weighted-voting, for which the best performing weights are received empirically. Thus, all possible ensemble models can be derived from the equation 2. Given $a_i^{ALiRAS^P}$, $a_i^{F^*}$, $p_i$ as the predicted probability of the class to be spoofed, then:

$$p_i^{ensemble} = [(weight1)(p_i^{F^*})] + [(weight2)(p_i^{ALiRAS^P})]. \tag{2}$$

Where $a_i^{F^*} = a_i^{deep-feature}$ and $p_i^{ensemble}$ is the final probability of audio clip $a_i$ to be spoofed, and weight1 + weight2 = 1. Figure 2 presents a toy example to clarify the steps of the ensemble model. b) Cost-efficient Ensemble Modeling: To optimize resource usage, including computational

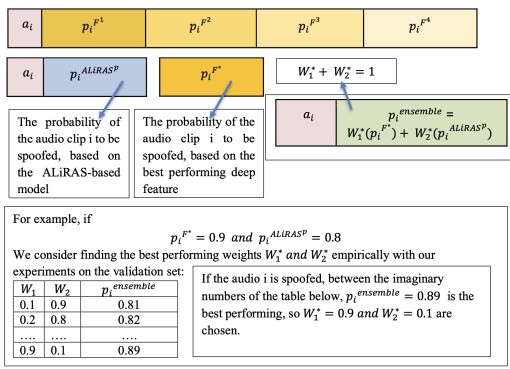

Figure 2: **Toy example of the ensemble model.** $p_i^{F^1}$ **refers to the probability of the audio clip i being spoofed when** $F_i^1$ **is given to a classifier as one of the input deep features (baselines). The best performing weights** ($W_1$ **and** $W_2$) **, which determine the contribution of deep features and ALiRAS-based model, are chosen empirically using the validation set.**

time, we implemented an alternative ensemble strategy. In this setup, the ALiRAS model is first used to classify audio clips as either genuine or spoofed. Based on empirical evaluation, a threshold of 0.55 was selected as the optimal decision boundary for the ALiRAS-based model. If a sample is classified as genuine (label = 0) by ALiRAS, it undergoes a second stage of classification using the baseline model. As a result, the computationally intensive feature extraction of large foundation models is performed only on a subset of the data, those initially labeled as genuine by ALiRAS— thereby reducing overall processing cost and time. Based on this ensemble setup, the decision function $f(x)$ for an audio sample $x$ is defined as depicted in equation 3.

$$f(x) = \begin{cases} \text{ALiRAS-MLP}(x), & \text{if ALiRAS-MLP}(x) = 1, \\ \text{Wav2Vec-XLSR-ResNet18}(x), & \text{if ALiRAS-MLP}(x) = 0 \end{cases} \tag{3}$$

**Explainability:** For extracting explainability or semantic meaning) of the predictions of the model, we use SHapley Additive exPlanations (SHAP) method Lundberg & Lee (2017). SHAP helps to understand individual predictions by assigning an importance score to each auto-labeled linguistic feature. It acts like a reverse engineering process, which interprets the decision of the model for the classification task. SHAP framework unifies multiple model interpretation methods strengthening its ability, and with ALiRAS , SHAP helps us interpreting the models' decisions. SHAP treats each auto-labeled linguistic feature like a player in a cooperative game. For every individual prediction, SHAP reveals how much each feature (player) contributed to the model's outputTrevisan (2022).

**Scalability:** Although foundation models offer automated feature extraction and are therefore inherently scalable, their slow processing speed limits their practical scalability. To address this limitation, we propose the expert-in-the-loop representation learning module that accelerates the feature extraction process by 31% on our large-scale dataset. Furthermore, we demonstrate that our approach is not constrained to manually extracted expert features like previous related studies Zahra Khanjani (2023); it generalizes effectively to large-scale scenarios through the aforementioned auto-labeling process.

**Effectiveness:** Now we explain the effectiveness evaluation methodology. One of the most popular metrics for anti-spoofing audio systems is Equal Error Rate (EER) which is also called Crossover Error Rate (CER) Conrad et al. (2017). The False Acceptance Rate (FAR) refers to the proportion of cases where the system incorrectly identifies an audio deepfake as genuine (equivalent to the False Negative Rate). Conversely, the False Rejection Rate (FRR) denotes the percentage of genuine samples that are wrongly classified as spoofed audio (equivalent to the False Positive Rate). The Equal Error Rate (EER), also referred to as the Crossover Error Rate (CER), is the point at which the FAR and FRR curves intersect or where they are very close Conrad et al. (2017) (finding the threshold). EER serves as an indicator of the overall performance of the ADD system. As system sensitivity increases, FRR tends to rise while FAR decreases, and vice versa Conrad et al. (2017). The mean of FAR and FRR at that threshold is usually preferred as EER: $EER = (FAR + FRR)/2$

Also, the final detection is a binary classification task, in an imbalance dataset, we use ROC AUC metric as well.

## EXPERIMENTS

In this section, we describe datasets used, implementation details, as well as results and analysis.

## DATASETS

Table 1 demonstrates the datasets used and their properties.

Table 1: Summary of the datasets utilized in this study.

| Dataset | Number of samples | Properties | References | Purpose |
|---|---|---|---|---|
| **Expert-labeled Dataset** | 840 audio samples | Attack types: Replay, Text-to-Speech, Voice Conversion. Balanced between spoofed and genuine samples. Linguistic features extracted manually by experts. English language; average duration: 4 seconds. 15% held out for evaluation/test | Reimao & Tzerpos (2019); Yamagishi et al. (2021); Wu et al. (2015); Kinnunen et al. (2017); Zahra Khanjani (2023); Kumar et al. (2019) | Root of ALiRAS; used for fine-tuning the auto-labeling model |
| **Large Scale Dataset** | 14,000 total (7,000 from ASVspoof 2021 DF evaluation set + 7,000 from ASVspoof 2019 LA training set) | Focus on AI-generated audio (Deepfake), Test set: 7,000 DF clips and Training set: 7,000 LA clips | Yamagishi et al. (2021; 2019) | Used for ADD research |

## FINDING APPROPRIATE EMBEDDINGS FOR ALiRAS

To auto-label the linguistic features we evaluated different front-ends including acoustic features as well as deep and foundation models. While the acoustic features, mentioned in the methodology, failed to capture the linguistic features with mostly low and less than chance accuracy, the deep features showed promising performance. Table 2 indicates different deep front-ends fine tuned with the manually extracted linguistic features, and their performance on the test set of the Expert-labeled Dataset.

As Table 2 indicates, VGGish shows better performance in auto-labeling the linguistic features. Therefore, the front-end chosen in ALiRAS is VGGish, as the model is fine tuned with the manually extracted Linguistic Features.

Table 2: **ROC AUC Scores for Different Front-ends Fine-tuned with Manually Extracted Linguistic Features to Label Audio Files with Binary Linguistic Features (Expert-labeled Dataset)**

| Front-end + Fine-tuned on | Average Test ROC AUC |
|---|---|
| Wav2vec-XLSR Babu et al. (2021) + manually extracted Linguistic Features | 0.57 |
| HuBERT Hsu et al. (2021) + manually extracted Linguistic Features | 0.59 |
| WavLM Chen et al. (2022) + manually extracted Linguistic Features | 0.59 |
| VGGish Hershey et al. (2017) + manually extracted Linguistic Features | **0.71** |

### EXPLAINABILITY

As we mentioned in the methodology section , we can use SHAP values in a reverse engineering process to investigate how much each attribute contributed to the model's decision for each individual clip. Figure 3b shows for this specific example of audio clips, the values of input features are 0.43 for audio quality, 0.51 for breath, 0.02 for anomalous pitch, and a base value is 0.81. Base value means the expected model output (i.e., the average prediction across the training data) before involving any features (initial prediction probability). These SHAP values represent the individual contributions of each feature in shifting the model's output from the base value: For example, audio quality (-0.0433) slightly decreased the spoof prediction (negative contribution) to the model's decision, slightly increasing the likelihood that the model predicts class as 0 (genuine). Breath for this particular audio clip increased the prediction towards Spoof label (1) by a value of (+0.0967), while pitch also increased the prediction towards class 1 by (+0.1367).

Figure 3a shows the mean of SHAP values for each feature across the Large-scale dataset. All three features are contributing meaningfully to the model's decision — their normalized (softmax) importance scores are fairly close indicating **Balanced Importance**; Audio Quality is the most influential, responsible for about 34% of the total impact. The differences between the importance scores are small, which implies that the ALiRAS-based model is not overly reliant on any single feature, and all three features are informative and relevant to the prediction task. The softmax function takes a list of values (like SHAP importance scores) and rescales them to a probability-like distribution, so can be especially useful for interpreting relative importance of each feature.

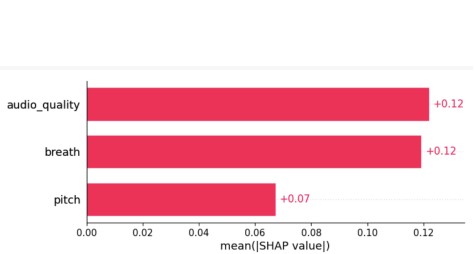

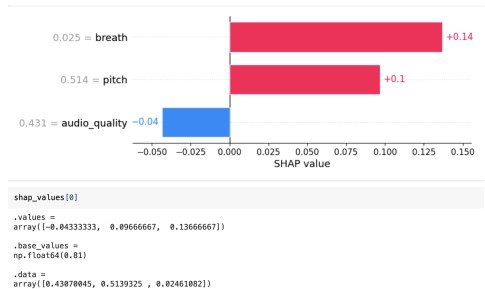

(a) Mean of SHAP values in Large-scale dataset    (b) SHAP values for an individual audio clip

Figure 3: **Explainability using SHAP values.** (a) Mean SHAP values across the large-scale dataset. (b) SHAP values for one random audio clip.

### SCALABILITY AND RESOURCE COMPARISON

Table 4 demonstrates time consumed for feature extraction when using a single GPU [2]. Even the fastest foundation model (HuBERT) is substantially more time-consuming than ALiRAS. ALiRAS completes extraction in approximately 15 seconds. This efficiency difference informed the design of our cost-effective ensemble. We aimed to leverage the speed of the ALiRAS method while simultaneously incorporating the benefits of the foundation models' architecture. As illustrated in Table 3, this approach resulted in a 31% reduction in extraction time without compromising the Equal Error Rate (EER) performance, as depicted in Table 5.

[2]NVIDA L40S GPU

Table 3: **Cost-efficient ensemble helped with the consumed time**

| Model | Processing Time ($\downarrow$) |
|---|---|
| XLSR-ResNet18 | 55:47:21 |
| ALiRAS-MLP\|XLSR-ResNet18 | 38:33:15 |
| HuBERT-ResNet18 | 29:54:10 |
| ALiRAS-MLP\|HuBERT-ResNet18 | 20:38:07 |
| WavLM-MLP | 43:59:13 |
| ALiRAS-MLP\|WavLM-MLP | 30:21:09 |

Table 4: **Resource Comparison for Different Representations**

| Representation | Number of GPUs Used | Extraction Time |
|---|---|---|
| VGGish | 1 | 44 minutes |
| Fastest Baseline (HuBERT) | 1 | 29 hours, 54 minutes, 10 seconds |
| ALiRAS | 0 | 15 seconds |

### EFFECTIVENESS

In this section, we evaluate the models in terms of their ability to ADD. As mentioned in Section
, as the ADD evaluation metrics, we consider accuracy for Expert-labeled Dataset, and ROC AUC
and EER for Large-scale dataset.

**Expert-labeled Dataset:** Expert-labeled Dataset was utilized to fine-tune the auto-labeling model
and output ALiRAS (auto-labeled linguistic features). As Table 2 indicates, manually extracted lin-
guistic features are provided in Expert-labeled Dataset.

**Large-scale Dataset:** Graph 4 shows how the ROC AUC score changes for different tried methods.
Our proposed ensemble method with the XLSR-based model has ROC AUC 0.84, and the cost-
efficient corresponding ensemble method has reduced this score to 0.78. However, both ensemble
setups offer the same EER metric (0.27) as depicted in Table 5. HuBERT achieves the lowest EER
(0.171), and incorporating it into the ensemble with ALiRAS does not lead to any increase in this
error rate. The cost-efficient ensemble with ALiRAS yields a slightly different EER (0.184) while
significantly reducing the resource and time demands of the feature extraction process as depicted
in Table 3.

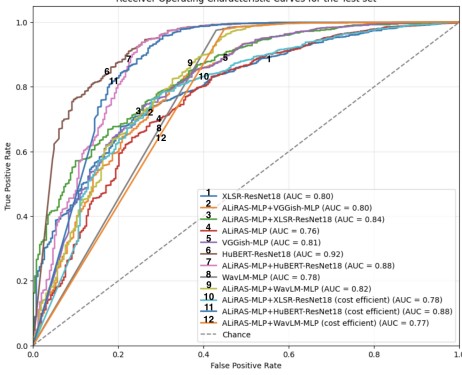

Figure 4: **ROC Comparison for Different Methods**

**Type of Attack Analysis:** We also explored how the methods work in terms of capturing differ-
ent types of audio deepfake (VC, TTS, VC-TTS and unknown attacks). For Large-scale dataset,
as depicted in Table 6, in ensemble modeling with XLSR, for all of the types of attacks, XLSR-
ResNet18—ALiRAS-MLP, has shown a stronger performance; only for VC-TTS is slightly lower
than for the XLSR-based model alone. For HuBERT and WavLM based models, the performance

Table 5: **EER (↓) Comparison for Different Methods on Large-scale Dataset**

| Model | EER (↓) |
|---|---|
| VGGish-MLP | 0.302 |
| ALiRAS-MLP | 0.319 |
| XLSR-ResNet18 | 0.400 |
| HuBERT-ResNet18 | 0.171 |
| WavLM-MLP | 0.277 |
| AliRAS-MLP—VGGish-MLP | 0.300 |
| AliRAS-MLP\|XLSR-ResNet18 | 0.274 |
| **AliRAS-MLP\|HuBERT-ResNet18** | **0.171** |
| AliRAS-MLP\|WavLM-MLP | 0.277 |
| AliRAS-MLP\|XLSR-ResNet18 (cost efficient) | 0.279 |
| **AliRAS-MLP\|HuBERT-ResNet18 (cost efficient)** | **0.184** |
| AliRAS-MLP\|WavLM-MLP (cost efficient) | 0.284 |

is the same across different types of attacks when ensemble with ALiRAS is applied. However, even for HuBERT and WavLM based models, the benefits of ALiRAS in terms of speeding feature extraction process by 31% and explainability still exist.

Table 6: **Type of attack analysis for Large-scale Dataset**

| Model | ROC AUC ↑ |
|---|---|
| **Type: TTS** | |
| VGGish-MLP | 0.799 |
| ALiRAS-MLP | 0.755 |
| XLSR-ResNet18 | 0.781 |
| ALiRAS-MLP \| VGGish-MLP | 0.794 |
| **ALiRAS-MLP \| XLSR-ResNet18** | **0.82** |
| HuBERT-ResNet18 / **ALiRAS-MLP \| HuBERT-ResNet18** | **0.926** |
| WavLM-MLP / **ALiRAS-MLP \| WavLM-MLP** | **0.781** |
| **Type: VC** | |
| VGGish-MLP | 0.661 |
| ALiRAS-MLP | 0.631 |
| XLSR-ResNet18 | 0.65 |
| ALiRAS-MLP \| VGGish-MLP | 0.659 |
| **ALiRAS-MLP \| XLSR-ResNet18** | **0.696** |
| HuBERT-ResNet18 / **ALiRAS-MLP \| HuBERT-ResNet18** | **0.793** |
| WavLM-MLP / **ALiRAS-MLP \| WavLM-MLP** | **0.694** |
| **Type: VC-TTS** | |
| VGGish-MLP | 0.821 |
| ALiRAS-MLP | 0.701 |
| XLSR-ResNet18 | 0.861 |
| ALiRAS-MLP \| VGGish-MLP | 0.794 |
| **ALiRAS-MLP \| XLSR-ResNet18** | **0.857** |
| HuBERT-ResNet18 / **ALiRAS-MLP \| HuBERT-ResNet18** | **0.908** |
| WavLM-MLP / **ALiRAS-MLP \| WavLM-MLP** | **0.785** |
| **Type: Unknown** | |
| VGGish-MLP | 0.822 |
| ALiRAS-MLP | 0.779 |
| XLSR-ResNet18 | 0.806 |
| ALiRAS-MLP \| VGGish-MLP | 0.813 |
| **ALiRAS-MLP \| XLSR-ResNet18** | **0.852** |
| HuBERT-ResNet18 / **ALiRAS-MLP \| HuBERT-ResNet18** | **0.931** |
| WavLM-MLP / **ALiRAS-MLP \| WavLM-MLP** | **0.785** |

## CONCLUSION

Current deepfake detection methods are increasingly challenged by rapid advancements in deepfake audio generation. In response, our work introduces a novel approach to deepfake speech detection by focusing on auto-labeling expert-in-the-loop representations with a phoneme-level view of speech. Experimental results demonstrate that our method consistently outperforms state-of-the-art baselines across multiple audio deepfake detection aspects: time (increasing the ability for real-time scenarios and scalability) and explainability (the ability to explain model decisions with tangible attributes and semantic meaning). For equal error rate (the ability to better detect spoofed samples), it either maintains or increases the performance of the state-of-the-art methods. More research will be important to explore the promising potential of this and similar expert-informed strategies.

ACKNOWLEDGMENTS

Audio datasets as well as all of the codes will be available through our GitHub repository for the camera ready version.

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
