# OpenReview forum: "Towards Multi-view, Explainable and Scalable Representation Learning for Spoofed Audio Detection"
_ICLR.cc/2026/Conference — Submitted to ICLR 2026_

### Official Review · Reviewer_ptqV · 2025-10-18

**Soundness:** 2
**Presentation:** 1
**Contribution:** 2
**Rating:** 2
**Confidence:** 4

**Summary:**

This paper proposes ALIRAS, a method for spoofed audio detection that aims to improve explainability, scalability, and effectiveness. The core idea is to augment deep learning features from foundation models (e.g., Wav2Vec-XLSR, HuBERT) with expert-informed linguistic cues (anomalous breath, pitch, and audio quality). To make this scalable, the authors train a lightweight VGGish-based model to auto-label these linguistic features on a large dataset after being trained on a small, manually-labeled set. The ALIRAS features are then combined with foundation model features through ensemble methods, including a cost-efficient cascade model to reduce computation time. The authors use SHAP to provide explanations for the model's predictions based on the linguistic features.

**Strengths:**

- The problem being addressed, creating explainable, scalable, and effective audio deepfake detectors, is timely and important for the AI community.
- The concept of a multi-view approach, combining low-level deep representations with high-level, human-interpretable linguistic features, is a conceptually interesting direction for improving model robustness and trustworthiness.

**Weaknesses:**

- Lack of Novelty: The proposed method is a straightforward combination of existing techniques. The use of expert-informed linguistic features for spoof detection builds directly on prior work cited by the authors (Zahra Khanjani, 2023). The "auto-labeling" component is a standard application of transfer learning or pseudo-labeling, and the ensemble techniques (weighted averaging and a cascaded filter) are rudimentary. The paper does not introduce any novel methodology to the field.
- Misleading Scalability Claims: The paper claims a 31% reduction in processing time as a scalability improvement. This is not a true algorithmic enhancement; it is a simple engineering trade-off where a faster, less accurate model is used to filter data for a slower, more powerful one. The comparison in Table 4, which pits the inference time of a tiny pre-trained model against the full feature extraction of a massive one, is an unfair comparison that exaggerates the benefit.
- Insufficient Data and Lack of Experimental Rigor: The study is not "large-scale" as claimed. The core ALIRAS model is fine-tuned on an "Expert-labeled Dataset" containing only 840 audio samples or an "Large Scale Dataset" containing only 7000 training samples, a size insufficient to guarantee a generalizable or robust model. The paper also omits crucial experimental details like data splits, training protocols, and hyperparameter tuning, which makes the results irreproducible.
- Limited Baseline Comparisons: The paper's evaluation is narrow, comparing only against three foundation model backbones (Wav2Vec-XLSR, HuBERT, and WavLM). It fails to benchmark against a wider array of state-of-the-art spoof detection systems, making it difficult to properly contextualize the performance and effectiveness of the proposed method.
- Poor Presentation Quality: The paper's presentation is below the expected standard for a top-tier conference. The manuscript suffers from unprofessional formatting and structural issues that impede clarity and credibility, making it difficult for readers to follow the methodology and results.

**Questions:**

Given the fundamental issues detailed in the Weaknesses section, I will forego listing minor clarification questions.

---

### Official Review · Reviewer_vRjh · 2025-10-27

**Soundness:** 2
**Presentation:** 2
**Contribution:** 2
**Rating:** 2
**Confidence:** 4

**Summary:**

The authors propose a multi-view representation learning method, ALIRAS, for spoofed audio detection. The method aims to enhance explainability and scalability by incorporating expert-defined linguistic features (respiration, pitch, audio quality) into deep learning models. The core of the approach involves fine-tuning a VGGish model on an expert-labeled dataset to create an auto-labeling system. This ALIRAS model is then integrated with larger foundation models (XLSR, HuBERT, WavLM) using ensemble strategies, with the authors claiming improvements in processing time and explainability while maintaining or enhancing detection performance.

**Strengths:**

- The paper addresses the critical and interconnected limitations of **explainability** and **scalability** in modern audio deepfake detection, which are major hurdles for deploying large foundation models.

- The concept of a **cost-efficient ensemble**—using a lightweight, expert-informed model (ALIRAS) as a pre-filter to reduce the computational load on heavier models—is a practical and novel architectural approach to improving scalability.

- The method offers a clear pathway to **explainability** by mapping model decisions back to tangible, human-understandable linguistic features (breath, pitch, quality) via SHAP analysis.

**Weaknesses:**

- **Questionable Effectiveness of the ALIRAS Method**: The paper's primary contribution, the ALIRAS module, shows questionable efficacy. As per Table 5, integrating ALIRAS with HuBERT-ResNet18 results in **no change** to the Equal Error Rate (EER) (0.171 vs 0.171). Similarly, the integration with VGGish-MLP shows only a negligible change (0.302 vs 0.300). This strongly suggests that the ALIRAS module provides little to no additive detection performance, directly undermining the paper's claims of effectiveness.
- **Insufficient Model Training Data**: The validity of the ALIRAS auto-labeling model, which is central to the entire method, is not convincing. As shown in Table 2, this model is fine-tuned on an Expert-labeled Dataset containing **only 840 samples**. This data volume is exceptionally small and very likely insufficient to train a model to reliably capture complex and subtle acoustic anomalies like "anomalous pitch" or "audio quality." The paper does not provide sufficient evidence that the fine-tuned VGGish model has truly learned these features rather than just overfitting to this tiny dataset.
- **Limited Benchmarking**: The evaluation presented in Table 5 is inadequate for a modern anti-spoofing paper. The authors did not compare their results against any current state-of-the-art (SOTA) countermeasures, such as **AASIST**, **XLSR_Mamba**.
- **Incomplete Dataset Evaluation**: The experimental setup on the primary dataset (ASVspoof 2021 DF, mentioned in Table 1) is not general. The authors only use a subset (7,000 samples) rather than the full, standard evaluation set. To make any claims of robustness, the method should have been tested on more diverse, "in-the-wild" datasets that include real-world distortions, such as **CodecFake** or the **In-The-Wild** dataset.
- **Presentation Errors**: The paper contains confusing errors. In Figure 1, there is a stray arrow originating from "FC(3)" that appears to be an artifact. In Table 5, the model "AliRAS-MLP-VGGish-MLP" uses a hyphen (-) which is inconsistent with the + notation used for all other ensembles (e.g., AliRAS-MLP + XLSR-ResNet18) and the methodology described in the text.

**Questions:**

- Given the extremely small Expert-labeled Dataset (840 samples), how can the authors be confident that the VGGish model (Table 2) learned generalizable representations of the three linguistic features? Was any k-fold cross-validation performed on this small dataset? What were the precision, recall, and F1-scores for detecting each of the three features individually?
- To experimentally verify the contribution of each linguistic feature in the ALIRAS method, have the authors conducted an ablation study? For instance, what is the performance (EER) of the ALIRAS-MLP model when trained and tested using only the 'breath' feature, only the 'pitch' feature, or only the 'audio quality' feature, respectively? Providing this experimental data would clearly demonstrate the necessity of all three features for the final model's performance.
- The performance of the "cost-efficient" ensemble is critically dependent on the 0.55 decision threshold selected for the ALIRAS-MLP model. How was this "optimal" threshold determined?

---

### Official Review · Reviewer_ZPBx · 2025-10-28

**Soundness:** 1
**Presentation:** 2
**Contribution:** 2
**Rating:** 2
**Confidence:** 3

**Summary:**

The paper tackles audio deepfake detection—classifying whether an audio sample is real or synthesized.
The main idea is to fuse two classifiers: (1) a classifier that predicts through three interpretable features (presence of breath, pitch anomaly, audio quality anomaly), and (2) a classifier based on self-supervised speech features.
The first classifier has the advantage of being more efficient (uses small networks to predict features) and more interpretable (predictions can be explained through these intermediary features).
The two classifiers are combined using an asymmetric rule: whenever the first classifier predicts "fake", the final prediction is "fake"; otherwise, the prediction of the second classifier is returned.
This approach is evaluated on 14,000 samples from ASVspoof 2019 and ASVspoof 2021.

**Strengths:**

- The paper addresses two important goals for practical deepfake detection: explainability and scalability.
- The simplicity of the approach is appreciated and makes it potentially practical to deploy.

**Weaknesses:**

- The evaluation is carried on a single dataset and there are no comparisons to prior work. For example, Martín Doñas et al. (2024) and Wang and Yamagishi (2024), amongst others, evaluate in the ASV19 → ASV21 setup. They obtain errors well as low as 1.16% EER, which are substantially better than the 17.1% EER reported in the paper.
- Given that the auto-labeling model is able to predict the interpretable features with only 71% AUC (Table 2), how can we trust it for the next stage (deepfake detection)? Note that this performance is also an upper-bound: on out-of-domain data (ASV19 or ASV21), I expect this value to be even lower.
- The point above raises concerns about the paper claim of explainability. How can we trust explanations based on "anomalous breath" or "pitch anomaly" when these features themselves are only ~70% accurate?
- The presentation of the paper could be improved. For example: parts of the methodological section includes ample discussion on prior work, which makes the paper hard to follow; the naming of the method is not always consistent, e.g., Table 5 explicitly names the "cost efficient" combinations, while in Table 3 this is left implicit; in Table 4 it's not immediately clear where the difference in time appears between VGGish and ALiRAS, which also relies on VGGish; the authors refer to the three features (breath, pitch, audio quality) as "linguistic", but these are rather phonetic or phonological features.
- The combinations of the two models biases the predictions towards the "fake" class (Eq. 3). Why not use the model's confidence as the gating criterion?

References:
- Martín Doñas, Juan Manuel, et al. "Exploring self-supervised embeddings and synthetic data augmentation for robust audio deepfake detection." *Interspeech*, 2024.
- Wang, Xin and Junichi Yamagishi, “Can large-scale vocoded spoofed data improve speech spoofing countermeasure with a self-supervised front end?” _ICASSP_, 2024.

**Questions:**

Questions:
- See weakness.
- How were the sizes of the auto-labeling networks been selected (L214–215)? Did you try training a similar sized VGGish network for the task of deepfake detection?
- What is the annotator inter-agreement of the "linguistic" annotations?

---

### Official Review · Reviewer_ASCz · 2025-10-31

**Soundness:** 2
**Presentation:** 2
**Contribution:** 2
**Rating:** 2
**Confidence:** 5

**Summary:**

This paper presents an Auto-labeled Linguistic Representations for Audio Spoofing Detection system, which is a multi-view, expert-in-the-loop framework for detecting AI-generated audio. The method combines foundation model features (e.g., Wav2Vec-XLSR, HuBERT, WavLM) with auto-labeled phonetic and phonological cues derived from expert linguistic annotations. Experiments show that ALiRAS achieves a 7% improvement in Equal Error Rate over XLSR-ResNet18 and a 31% reduction in processing time.

**Strengths:**

1. The paper bridges sociolinguistics and deep learning, introducing an automatic labeling process that scales previous human-expert-based approaches.

2. The authors build upon and extend the previous expert-in-the-loop framework, showing continuity and advancement in explainable ADD design.

**Weaknesses:**

1. **The paper’s organization could be significantly improved.** In particular, the Introduction section contains an extensive discussion of prior work that would be more appropriately placed in a Related Work section. Additionally, the citation style and formatting are inconsistent and do not adhere to standard conventions, which disrupts the reading flow. The roadmap paragraph also lacks clear section references (e.g., section numbers), making it harder for readers to follow the paper’s structure. Overall, these issues reduce the clarity and readability of the paper.

2. While the integration of linguistic auto-labeling is novel, the underlying architecture (ensemble of MLP + ResNet18 with existing embeddings) may be seen as incremental rather than fundamentally novel.

3. The linguistic features (breath, pitch, audio quality) are few and coarse-grained; it remains unclear whether such limited cues can generalize to diverse languages, speakers, or unseen generation methods.

4. All linguistic labeling and evaluation are English-based, using ASVspoof subsets. No cross-lingual or real-world noisy evaluation is provided to support the “scalable” or “generalizable” claims.

**Questions:**

1. How well do the auto-labeled linguistic features generalize to unseen datasets or non-English speech without expert retraining?

2. Did you perform any human evaluation (e.g., linguist or user study) to verify that SHAP explanations correspond to meaningful phonetic phenomena?

3. Have the authors conducted ablation studies to investigate how much each component (auto-labeling, cost-efficient ensemble, linguistic feature type) contributes to the overall performance gain?

---

### Meta-Review · Area_Chair_FPWL · 2026-01-05

**Summary:**

This submission received consistent ratings of rejection, and the authors did not provide any rebuttal or revision. Therefore, my decision is Rejection.

Here is a summary of major weaknesses:
- Presentations need to be more precise and accurate for clarity
- insufficient model training and data
- insffuciient comparisons to existing works

**Reviewer Scores:**

NA

---

### Decision · Program_Chairs · 2026-01-26

Reject